# GEOMETRY OF PROGRAM SYNTHESIS

## ABSTRACT

We present a new perspective on program synthesis in which programs may be identified with singularities of analytic functions. As an example, Turing machines are synthesised from input-output examples by propagating uncertainty through a smooth relaxation of a universal Turing machine. The posterior distribution over weights is approximated using Markov chain Monte Carlo and bounds on the generalisation error of these models is estimated using the real log canonical threshold, a geometric invariant from singular learning theory.

## 1 INTRODUCTION

The idea of program synthesis dates back to the birth of modern computation itself (Turing, 1948) and is recognised as one of the most important open problems in computer science (Gulwani et al., 2017). However, there appear to be serious obstacles to synthesising programs by gradient descent at scale (Neelakantan et al., 2016; Kaiser & Sutskever, 2016; Bunel et al., 2016; Gaunt et al., 2016; Evans & Grefenstette, 2018; Chen et al., 2018) and these problems suggest that it would be appropriate to make a fundamental study of the geometry of loss surfaces in program synthesis, since this geometry determines the learning process. To that end, in this paper we explain a new point of view on program synthesis using the singular learning theory of Watanabe (2009) and the smooth relaxation of Turing machines from Clift & Murfet (2018).

In broad strokes this new geometric point of view on program synthesis says:

- **Programs to be synthesised are singularities of analytic functions**. If $U \subseteq \mathbb{R}^d$ is open and $K : U \longrightarrow \mathbb{R}$ is analytic, then $x \in U$ is a *critical point* of $K$ if $\nabla K(x) = 0$ and a *singularity* of the function $K$ if it is a critical point where $K(x) = 0$.

- **The Kolmogorov complexity of a program is related to a geometric invariant** of the associated singularity called the Real Log Canonical Threshold (RLCT). This invariant controls both the generalisation error and the learning process, and is therefore an appropriate measure of "complexity" in continuous program synthesis. See Section 3.

- **The geometry has concrete practical implications**. For example, a MCMC-based approach to program synthesis will find, with high probability, a solution that is of low complexity (if it finds a solution at all). We sketch a novel point of view on the problem of "bad local minima" (Gaunt et al., 2016) based on these ideas. See Section 4.

We demonstrate all of these principles in experiments with toy examples of synthesis problems.

**Program synthesis as inference.** We use Turing machines, but *mutatis mutandis* everything applies to other programming languages. Let $T$ be a Turing machine with tape alphabet $\Sigma$ and set of states $Q$ and assume that on any input $x \in \Sigma^*$ the machine eventually halts with output $T(x) \in \Sigma^*$. Then to the machine $T$ we may associate the set $\{(x, T(x))\}_{x \in \Sigma^*} \subseteq \Sigma^* \times \Sigma^*$. Program synthesis is the study of the inverse problem: given a subset of $\Sigma^* \times \Sigma^*$ we would like to determine (if possible) a Turing machine which computes the given outputs on the given inputs.

If we presume given a probability distribution $q(x)$ on $\Sigma^*$ then we can formulate this as a problem of statistical inference: given a probability distribution $q(x, y)$ on $\Sigma^* \times \Sigma^*$ determine the most likely machine producing the observed distribution $q(x, y) = q(y|x)q(x)$. If we fix a universal Turing machine $\mathcal{U}$ then Turing machines can be parametrised by codes $w \in W^{code}$ with $\mathcal{U}(x, w) = T(x)$ for all $x \in \Sigma^*$. We let $p(y|x, w)$ denote the probability of $\mathcal{U}(x, w) = y$ (which is either zero or one)

so that solutions to the synthesis problem are in bijection with the zeros of the Kullback-Leibler divergence between the true distribution and the model

$$K(w) = \int \int q(y|x)q(x) \log \frac{q(y|x)}{p(y|x, w)} dxdy \,. \tag{1}$$

So far this is just a trivial rephrasing of the combinatorial optimisation problem of finding a Turing machine $T$ with $T(x) = y$ for all $(x, y)$ with $q(x, y) > 0$.

**Smooth relaxation.** One approach is to seek a smooth relaxation of the synthesis problem consisting of an analytic manifold $W \supseteq W^{code}$ and an extension of $K$ to an analytic function $K : W \longrightarrow \mathbb{R}$ so that we can search for the zeros of $K$ using gradient descent. Perhaps the most natural way to construct such a smooth relaxation is to take $W$ to be a space of probability distributions over $W^{code}$ and prescribe a model $p(y|x, w)$ for propagating uncertainty about codes to uncertainty about outputs (Gaunt et al., 2016; Evans & Grefenstette, 2018). The particular model we choose is based on the semantics of linear logic (Clift & Murfet, 2018). Supposing that such a smooth relaxation has been chosen together with a prior $\varphi(w)$ over $W$, smooth program synthesis becomes the study of the statistical learning theory of the triple $(p, q, \varphi)$.

There are perhaps two primary reasons to consider the smooth relaxation. Firstly, one might hope that stochastic gradient descent or techniques like Markov chain Monte Carlo will be effective means of solving the original combinatorial optimisation problem. This is not a new idea (Gulwani et al., 2017, §6) but so far its effectiveness for large programs has not been proven. Independently, one might hope to find powerful new mathematical ideas that apply to the relaxed problem and shed light on the nature of program synthesis. This is the purpose of the present paper.

**Singular learning theory.** We denote by $W_0 = \{w \in W \,|\, K(w) = 0\}$ so that

$$W_0 \cap W^{code} \subseteq W_0 \subseteq W \tag{2}$$

where $W_0 \cap W^{code}$ is the discrete set of solutions to the original synthesis problem. We refer to these as the *classical solutions*. As the vanishing locus of an analytic function, $W_0$ is an analytic space over $\mathbb{R}$ (Hironaka, 1964, §0.1), (Griffith & Harris, 1978) and it is interesting to study the geometry of this space near the classical solutions. Since $K$ is a Kullback-Leibler divergence it is non-negative and so it not only vanishes on $W_0$ but $\nabla K$ also vanishes, hence every point of $W_0$ is a singular point.

Beyond this the geometry of $W_0$ depends on the particular model $p(y|x, w)$ that has been chosen, but some aspects are universal: the nature of program synthesis means that typically $W_0$ is an extended object (i.e. it contains points other than the classical solutions) and the Hessian matrix of second order partial derivatives of $K$ at a classical solution is not invertible - that is, the classical solutions are *degenerate* critical points of $K$. This means that singularity theory is the appropriate branch of mathematics for studying the geometry of $W_0$ near a classical solution. It also means that the Fisher information matrix

$$I(w)_{ij} = \int \int \frac{\partial}{\partial w_i} \big[ \log p(y|x, w) \big] \frac{\partial}{\partial w_j} \big[ \log p(y|x, w) \big] q(y|x)q(x)dxdy,$$

is degenerate at a classical solution, so that the appropriate branch of statistical learning theory is singular learning theory (Watanabe, 2007; 2009). For an introduction to singular learning theory in the context of deep learning see (Murfet et al., 2020).

Broadly speaking the contribution of this paper is to realise program synthesis within the framework of singular learning theory, at both a theoretical and an experimental level. In more detail the contents of the paper are:

- We define a *staged pseudo-UTM* (Appendix E) which is well-suited to experiments with the ideas discussed above. Propagating uncertainty about the code through this UTM using the ideas of (Clift & Murfet, 2018) defines a triple $(p, q, \varphi)$ associated to a synthesis problem. This formally embeds program synthesis within singular learning theory.

- We realise this embedding in code by providing an implementation in PyTorch of this propagation of uncertainty through a UTM. Using the No-U-Turn variant of MCMC (Hoffman & Gelman, 2014) we can approximate the Bayesian posterior of any program synthesis problem (of course in practice we are limited by computational constraints in doing so).

- We explain how the real log canonical threshold (a geometric invariant) is related to Kolmogorov complexity (Section 3).

- We give a simple example (Appendix C) in which $W_0$ contains the set of classical solutions as a proper subset and every point of $W_0$ is a degenerate critical point of $K$.

- For two simple synthesis problems `detectA` and `parityCheck` we demonstrate all of the above, using MCMC to approximate the Bayesian posterior and theorems from Watanabe (2013) to estimate the RLCT (Section 5). We discuss how $W_0$ is an extended object and how the RLCT relates to the local dimension of $W_0$ near a classical solution.

### RELATED WORK

The idea of synthesising Turing machines can be traced back to the work of Solomonoff on inductive inference (Solomonoff, 1964). A more explicit form of the problem was given in Biermann (1972) who proposed an algorithmic method. Machine learning based approaches appear in Schmidhuber (1997) and Hutter (2004), which pay particular attention to model complexity, and Gaunt et al. (2016) and Freer et al. (2014), the latter using the notion of "universal probabilistic Turing machine" (De Leeuw et al., 1956). A different probabilistic extension of a universal Turing machine was introduced in Clift & Murfet (2018) via linear logic. Studies of the singular geometry of learning models go back to Amari et al. (2003) and notably, the extensive work of Watanabe (2007; 2009).

## 2   TURING MACHINE SYNTHESIS AS SINGULAR LEARNING

All known approaches to program synthesis can be formulated in terms of a singular learning problem. Singular learning theory is the extension of statistical learning theory to account for the fact that the set of learned parameters $W_0$ has the structure of an analytic space as opposed to an analytic manifold (Watanabe, 2007; 2009). It is organised around triples $(p, q, \varphi)$ consisting of a class of models $\{p(y|x,w) : w \in W\}$, a true distribution $q(y|x)$ and a prior $\varphi$ on $W$.

In our approach we fix a Universal Turing Machine (UTM), denoted $\mathcal{U}$, with a *description tape* (which specifies the code of the Turing machine to be executed), a *work tape* (simulating the tape of that Turing machine during its operation) and a *state tape* (simulating the state of that Turing machine). The general statistical learning problem that can be formulated using $\mathcal{U}$ is the following: given some initial string $x$ on the work tape, predict the state of the simulated machine and the contents of the work tape after some specified number of steps (Clift & Murfet, 2018, §7.1). For simplicity, in this paper we consider models that only predict the final state; the necessary modifications in the general case are routine. We also assume that $W$ parametrises Turing machines whose tape alphabet $\Sigma$ and set of states $Q$ have been encoded by individual symbols in the tape alphabet of $\mathcal{U}$. Hence $\mathcal{U}$ is actually what we call a *pseudo-UTM* (see Appendix E). Again, treating the general case is routine and for the present purposes only introduces uninteresting complexity.

Let $\Sigma$ denote the tape alphabet of the simulated machine, $Q$ the set of states and let $L, S, R$ stand for *left*, *stay* and *right*, the possible motions of the Turing machine head. We assume that $|Q| > 1$ since otherwise the synthesis problem is trivial. The set of ordinary codes $W^{code}$ for a Turing machine sits inside a compact space of probability distributions $W$ over codes

$$W^{code} := \prod_{\sigma,q} \Sigma \times Q \times \{L, S, R\} \subseteq \prod_{\sigma,q} \Delta\Sigma \times \Delta Q \times \Delta\{L, S, R\} =: W \qquad (3)$$

where $\Delta X$ denotes the set of probability distributions over a set $X$, see (8), and the product is over pairs $(\sigma, q) \in \Sigma \times Q$.[1] For example the point $\{(\sigma', q', d)\}_{\sigma,q} \in W^{code}$ encodes the machine which when it reads $\sigma$ under the head in state $q$ writes $\sigma'$, transitions into state $q'$ and moves in direction $d$. Given $w \in W^{code}$ let $\text{step}^t(x, w) \in Q$ denote the contents of the state tape of $\mathcal{U}$ after $t$ timesteps (of the simulated machine) when the work tape is initialised with $x$ and the description tape with $w$.

---

[1]The space $W$ of parameters is clearly semi-analytic, that is, it is cut out of $\mathbb{R}^d$ for some $d$ by the vanishing $f_1(x) = \cdots = f_r(x) = 0$ of finitely many analytic functions on open subsets of $\mathbb{R}^d$ together with finitely many inequalities $g_1(x) \geq 0, \ldots, g_s(x) \geq 0$ where the $g_j(x)$ are analytic. In fact $W$ is semi-algebraic, since the $f_i$ and $g_j$ may all be chosen to be polynomial functions.

There is a principled extension of this operation of $\mathcal{U}$ to a smooth function

$$\Delta\,\mathrm{step}^t : \Sigma^* \times W \longrightarrow \Delta Q \qquad (4)$$

which propagates uncertainty about the symbols on the description tape to uncertainty about the final state and we refer to this extension as the *smooth relaxation* of $\mathcal{U}$. The details are given in Appendix F but at an informal level the idea behind the relaxation is easy to understand: to sample from $\Delta\,\mathrm{step}^t(x,w)$ we run $\mathcal{U}$ to simulate $t$ timesteps in such a way that whenever the UTM needs to "look at" an entry on the description tape we sample from the corresponding distribution specified by $w$.[2] The significance of the particular smooth relaxation that we use is that its derivatives have a logical interpretation (Clift & Murfet, 2018, §7.1).

The class of models that we consider is

$$p(y|x,w) = \Delta\,\mathrm{step}^t(x,w) \qquad (5)$$

where $t$ is fixed for simplicity in this paper. More generally we could also view $x$ as consisting of a sequence and a timeout, as is done in (Clift & Murfet, 2018, §7.1). The construction of this model is summarised in Figure 1.

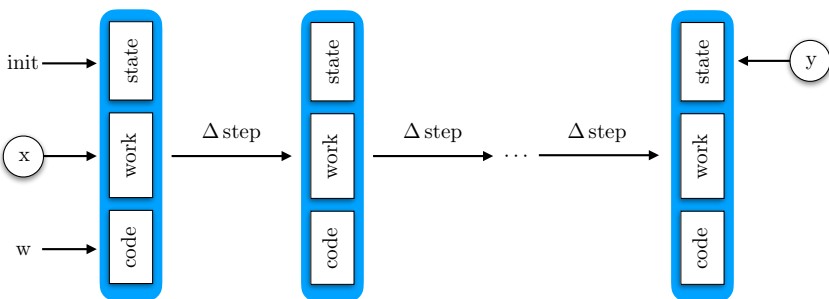

Figure 1: The state of $\mathcal{U}$ is represented by the state of the work tape, state tape and description (code) tape. The work tape is initialised with a sequence $x \in \Sigma^*$, the code tape with $w \in W$ and the state tape with some standard initial state, the smooth relaxation $\Delta\,\mathrm{step}$ of the pseudo-UTM is run for $t$ steps and the final probability distribution over states is $y$.

**Definition 2.1 (Synthesis problem).** A *synthesis problem* for $\mathcal{U}$ consists of a probability distribution $q(x,y)$ over $\Sigma^* \times Q$. We say that the synthesis problem is *deterministic* if there is $f : \Sigma^* \longrightarrow Q$ such that $q(y = f(x)|x) = 1$ for all $x \in \Sigma^*$.

**Definition 2.2.** The triple $(p, q, \varphi)$ associated to a synthesis problem is the model $p$ of (5) together with the true distribution $q$ and uniform prior $\varphi$ on the parameter space $W$. The Kullback-Leibler function $K(w)$ of the synthesis problem is defined by (1) and a *solution* to the synthesis problem is a point of $W_0$. A *classical solution* is a point of $W_0 \cap W^{code}$.

As $\Delta\,\mathrm{step}^t$ is a polynomial function, $K$ is analytic and so $W_0$ is a semi-analytic space (it is cut out of the semi-analytic space $W$ by the vanishing of $K$). If the synthesis problem is deterministic and $q(x)$ is uniform on some finite subset of $\Sigma^*$ then $W_0$ is semi-algebraic (it is cut out of $W$ by polynomial equations) and all solutions lie at the boundary of the parameter space $W$ (Appendix D). However in general $W_0$ is only semi-analytic and intersects the interior of $W$ (Example C.2). We assume that $q(y|x)$ is *realisable* that is, there exists $w_0 \in W$ with $q(y|x) = p(y|x, w_0)$.

A triple $(p, q, \varphi)$ is *regular* if the model is identifiable, ie. for all inputs $x \in \mathbb{R}^n$, the map sending $w$ to the conditional probability distribution $p(y|x, w)$ is one-to-one, and the Fisher information matrix is non-degenerate. Otherwise, the learning machine is *strictly singular* (Watanabe, 2009, §1.2.1). Triples arising from synthesis problems are typically singular: in Example 2.5 below we show an explicit example where multiple parameters $w$ determine the same model, and in Example C.2 we give an example where the Hessian of $K$ is degenerate everywhere on $W_0$ (Watanabe, 2009, §1.1.3).

---

[2]Noting that this sampling procedure is repeated *every time* the UTM looks at a given entry.

**Remark 2.3.** Non-deterministic synthesis problems arise naturally in various contexts, for example in the fitting of algorithms to the behaviour of deep reinforcement learning agents. Suppose an agent is acting in an environment with starting states encoded by $x \in \Sigma^*$ and possible episode end states by $y \in Q$. Even if the optimal policy is known to determine a computable function $\Sigma^* \longrightarrow Q$ the statistics of the observed behaviour after finite training time will only provide a function $\Sigma^* \longrightarrow \Delta Q$ and if we wish to fit algorithms to behaviour it makes sense to deal with this uncertainty directly.

**Definition 2.4.** Let $(p, q, \varphi)$ be the triple associated to a synthesis problem. The *Real Log Canonical Threshold* (RLCT) $\lambda$ of the synthesis problem is defined so that $-\lambda$ is the largest pole of the meromorphic extension (Atiyah, 1970) of the zeta function $\zeta(z) = \int K(w)^z \varphi(w) dw$.

The more singular the analytic space $W_0$ of solutions is, the smaller the RLCT. One way to think of the RLCT is as a count of the effective number of parameters near $W_0$ (Murfet et al., 2020, §4). In Section 3 we relate the RLCT to Kolmogorov complexity and in Section 5 we estimate the RLCT of the synthesis problem `detectA` given below, using the method explained in Appendix A.

**Example 2.5 (`detectA`).** The deterministic synthesis problem `detectA` has $\Sigma = \{\square, A, B\}$, $Q = \{\text{reject}, \text{accept}\}$ and $q(y|x)$ is determined by the function taking in a string $x$ of $A$'s and $B$'s and returning the state $\text{accept}$ if the string contains an $A$ and state $\text{reject}$ otherwise. The conditional true distribution $q(y|x)$ is realisable because this function is computed by a Turing machine.

Two solutions are shown in Figure 2. On the left is a parameter $w_l \in W_0 \setminus W^{code}$ and on the right is $w_r \in W_0 \cap W^{code}$. Varying the distributions in $w_l$ that have nonzero entropy we obtain a submanifold $V \subseteq W_0$ containing $w_l$ of dimension 14. This leads by (Watanabe, 2009, Remark 7.3) to a bound on the RLCT of $\lambda \leq \frac{1}{2}(30 - 14) = 8$ which is consistent with the experimental results in Table 1. This highlights that solutions need not lie at vertices of the probability simplex, and $W_0$ may contain a high-dimensional submanifold around a given classical solution.

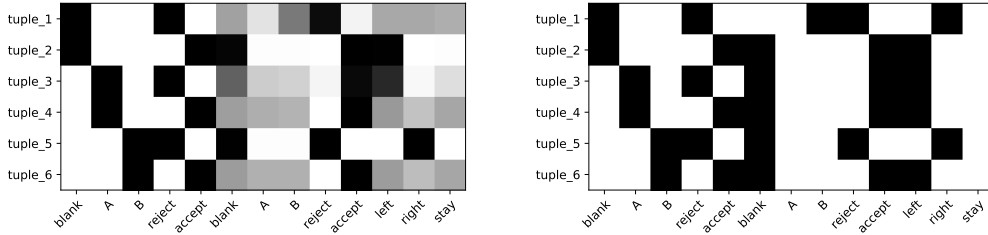

Figure 2: Visualisation of two solutions for the synthesis problem `detectA` .

## 2.1 THE SYNTHESIS PROCESS

Synthesis is a *problem* because we do not assume that the true distribution is known: for example, if $q(y|x)$ is deterministic and the associated function is $f : \Sigma^* \longrightarrow Q$, we assume that some example pairs $(x, f(x))$ are known but no general algorithm for computing $f$ is known (if it were, synthesis would have already been performed). In practice synthesis starts with a sample $D_n = \{(x_i, y_i)\}_{i=1}^n$ from $q(x, y)$ with associated empirical Kullback-Leibler distance

$$K_n(w) = \frac{1}{n} \sum_{i=1}^{n} \log \frac{q(y_i|x_i)}{p(y_i|x_i, w)} . \tag{6}$$

If the synthesis problem is deterministic and $u \in W^{code}$ then $K_n(u) = 0$ if and only if $u$ explains the data in the sense that $\text{step}^t(x_i, u) = y_i$ for $1 \leq i \leq n$. We now review two natural ways of finding such solutions in the context of machine learning.

**Synthesis by stochastic gradient descent (SGD).** The first approach is to view the process of program synthesis as stochastic gradient descent for the function $K : W \longrightarrow \mathbb{R}$. We view $D_n$ as a large training set and further sample subsets $D_m$ with $m \ll n$ and compute $\nabla K_m$ to take gradient descent steps $w_{i+1} = w_i - \eta \nabla K_m(w_i)$ for some learning rate $\eta$. Stochastic gradient descent has the advantage (in principle) of scaling to high-dimensional parameter spaces $W$, but in practice it is challenging to use gradient descent to find points of $W_0$ (Gaunt et al., 2016).

**Synthesis by sampling.** The second approach is to consider the Bayesian posterior associated to the synthesis problem, which can be viewed as an update on the prior distribution $\varphi$ after seeing $D_n$

$$p(w|D_n) = \frac{p(D_n|w)p(w)}{p(D_n)} = \frac{1}{Z_n}\varphi(w)\prod_{i=1}^{n}p(y_i|x_i, w) = \frac{1}{Z_n^0}\exp\{-nK_n(w) + \log\varphi(w)\}$$

where $Z_n^0 = \int \varphi(w)\exp(-nK_n(w))dw$. If $n$ is large the posterior distribution concentrates around solutions $w \in W_0$ and so sampling from the posterior will tend to produce machines that are (nearly) solutions. The gold standard sampling is Markov Chain Monte Carlo (MCMC). Scaling MCMC to where $W$ is high-dimensional is a challenging task with many attempts to bridge the gap with SGD (Welling & Teh, 2011; Chen et al., 2014; Ding et al., 2014; Zhang et al., 2020). Nonetheless in simple cases we demonstrate experimentally in Section 5 that machines may be synthesised by using MCMC to sample from the posterior.

## 3 COMPLEXITY OF PROGRAMS

Every Turing machine is the solution of a deterministic synthesis problem, so Section 2 associates to any Turing machine a singularity of a semi-analytic space $W_0$. To indicate that this connection is not vacuous, we sketch how the complexity of a program is related to the real log canonical threshold of a singularity. A more detailed discussion will appear elsewhere.

Let $q(x, y)$ be a deterministic synthesis problem for $\mathcal{U}$ which only involves input sequences in some restricted alphabet $\Sigma_{input}$, that is, $q(x) = 0$ if $x \notin (\Sigma_{input})^*$. Let $D_n$ be sampled from $q(x, y)$ and let $u, v \in W^{code} \cap W_0$ be two explanations for the sample in the sense that $K_n(u) = K_n(v) = 0$. Which explanation for the data should we prefer? The classical answer based on Occam's razor (Solomonoff, 1964) is that we should prefer the shorter program, that is, the one using the fewest states and symbols.

Set $N = |\Sigma|$ and $M = |Q|$. Any Turing machine $T$ using $N' \leq N$ symbols and $M' \leq M$ states has a code for $\mathcal{U}$ of length $cM'N'$ where $c$ is a constant. We assume that $\Sigma_{input}$ is included in the tape alphabet of $T$ so that $N' \geq |\Sigma_{input}|$ and define the *Kolmogorov complexity* of $q$ with respect to $\mathcal{U}$ to be the infimum $\mathfrak{c}(q)$ of $M'N'$ over Turing machines $T$ that give classical solutions for $q$.

Let $\lambda$ be the RLCT of the triple $(p, q, \varphi)$ associated to the synthesis problem (Definition 2.4).

**Theorem 3.1.** $\lambda \leq \frac{1}{2}(M + N)\mathfrak{c}(q)$.

*Proof.* Let $u \in W^{code} \cap W_0$ be the code of a Turing machine realising the infimum in the definition of the Kolmogorov complexity and suppose that this machine only uses symbols in $\Sigma'$ and states in $Q'$ with $N' = |\Sigma'|$ and $M' = |Q'|$. The time evolution of the staged pseudo-UTM $\mathcal{U}$ simulating $u$ on $x \in \Sigma_{input}^*$ is independent of the entries on the description tape that belong to tuples of the form $(\sigma, q, ?, ?, ?)$ with $(\sigma, q) \notin \Sigma' \times Q'$. Let $V \subseteq W$ be the submanifold of points which agree with $u$ on all tuples with $(\sigma, q) \in \Sigma' \times Q'$ and are otherwise free. Then $u \in V \subseteq W_0$ and $\text{codim}(V) = M'N'(M + N)$ and by (Watanabe, 2009, Theorem 7.3) we have $\lambda \leq \frac{1}{2}\text{codim}(V)$. □

**Remark 3.2.** The Kolmogorov complexity depends only on the *number* of symbols and states used. The RLCT is a more refined invariant since it also depends on *how* each symbol and state is used (Clift & Murfet, 2018, Remark 7.8) as this affects the polynomials defining $W_0$ (see Appendix D).

## 4 PRACTICAL IMPLICATIONS

Using singular learning theory we have explained how programs to be synthesised are singularities of analytic functions, and how the Kolmogorov complexity of a program bounds the RLCT of the associated singularity. We now sketch some practical insights that follow from this point of view.

**Synthesis minimises the free energy:** the sampling-based approach to synthesis (Section 2.1) aims to approximate, via MCMC, sampling from the Bayesian posterior for the triple $(p, q, \varphi)$ associated to a synthesis problem. To understand the behaviour of these Markov chains we follow the asymptotic analysis of (Watanabe, 2009, Section 7.6). If we cover $W$ by small closed balls $V_\alpha$ around

points $w_\alpha$ then we can compute the probability that a sample comes from $V_\alpha$ by

$$p_\alpha = \frac{1}{Z_0} \int_{V_\alpha} e^{-nK_n(w)} \varphi(w) dw$$

and if $n$ is sufficiently large this is proportional to $e^{-f_\alpha}$ where the quantity

$$f_\alpha = K_\alpha n + \lambda_\alpha \log(n)$$

is called the *free energy*. Here $K_\alpha$ is the smallest value of the Kullback-Leibler divergence $K$ on $V_\alpha$ and $\lambda_\alpha$ is the RLCT of the set $W_{K_\alpha} \cap V_\alpha$ where $W_c = \{w \in W \mid K(w) = c\}$ is a level set of $K$. The Markov chains used to generate approximate samples from the posterior are attempting to minimise the free energy, which involves a tradeoff between the energy $K_\alpha n$ and the entropy $\lambda_\alpha \log(n)$.

**Why synthesis gets stuck:** the kind of local minimum of the free energy that we *want* the synthesis process to find are solutions $w_\alpha \in W_0$ where $\lambda_\alpha$ is minimal. By Section 3 one may think of these points as the "lowest complexity" solutions. However it is possible that there are other local minima of the free energy. Indeed, there may be local minima where the free energy is *lower than the free energy at any solution* since at finite $n$ it is possible to tradeoff an increase in $K_\alpha$ against a decrease in the RLCT $\lambda_\alpha$. In practice, the existence of such "siren minima" of the free energy may manifest itself as regions where the synthesis process gets stuck and fails to converge to a solution. In such a region $K_\alpha n + \lambda_\alpha \log(n) < \lambda \log(n)$ where $\lambda$ is the RLCT of the synthesis problem. In practice it has been observed that program synthesis by gradient descent often fails for complex problems in the sense that it fails to converge to a solution (Gaunt et al., 2016). While synthesis by SGD and sampling are different, it is a reasonable hypothesis that these siren minima are a significant contributing factor in both cases.

**Can we avoid siren minima?** If we let $\lambda_c$ denote the RLCT of the level set $W_c$ then siren minima of the free energy will be impossible at a given value of $n$ and $c$ as long as $\lambda_c \geq \lambda - c\frac{n}{\log(n)}$. Recall that the more singular $W_c$ is the lower the RLCT, so this lower bound says that the level sets should not become too singular too quickly as $c$ increases. At any given value of $n$ there is a "siren free" region in the range $c \geq \frac{\lambda \log(n)}{n}$ since the RLCT is non-negative (Figure 3). Thus the learning process will be more reliable the smaller $\frac{\lambda \log(n)}{n}$ is. This can arranged either by increasing $n$ (providing more examples) or decreasing $\lambda$.

While the RLCT is determined by the synthesis problem, it is possible to change its value by changing the structure of the UTM $\mathcal{U}$. As we have defined it $\mathcal{U}$ is a "simulation type" UTM, but one could for example add special states such that if a code specifies a transition into that state a series of steps is executed by the UTM (i.e. a subroutine). This amounts to specifying codes in a higher level programming language. Hence one of the practical insights that can be derived from the geometric point of view on program synthesis is that varying this language is a natural way to engineer the singularities of the level sets of $K$, which according to singular learning theory has direct implications for the learning process.

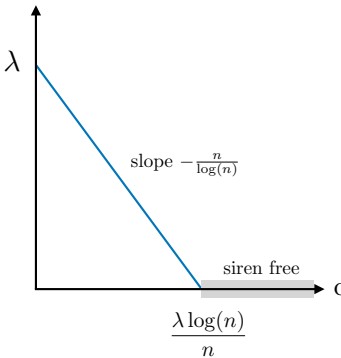

Figure 3: Level sets above the cutoff cannot contain siren local minima of the free energy.

## 5 EXPERIMENTS

We estimate the RLCT for the triples $(p, q, \varphi)$ associated to the synthesis problems `detectA` (Example 2.5) and `parityCheck`. Hyperparameters of the various machines are contained in Table 3 of Appendix B. The true distribution $q(x)$ is defined as follows: we fix a minimum and maximum sequence length $a \leq b$ and to sample $x \sim q(x)$ we first sample a length $l$ uniformly from $[a, b]$ and then uniformly sample $x$ from $\{A, B\}^l$.

We perform MCMC on the weight vector for the model class $\{p(y|x, w) : w \in W\}$ where $w$ is represented in our PyTorch implementation by three tensors of shape $\{[L, n_i]\}_{1 \leq i \leq 3}$ where $L$ is the number of tuples in the description tape of the TM being simulated and $\{n_i\}$ are the number of symbols, states and directions respectively. A *direct simulation* of the UTM is used for all experiments to improve computational efficiency (Appendix G). We generate, for each inverse temperature $\beta$ and dataset $D_n$, a Markov chain via the No-U-turn sampler from Hoffman & Gelman (2014). We use the standard uniform distribution as our prior $\varphi$.

| Max-length | Temperature | RLCT | Std | R squared |
|---|---|---|---|---|
| 7 | $\log(500)$ | 8.089205 | 3.524719 | 0.965384 |
| 7 | $\log(1000)$ | 6.533362 | 2.094278 | 0.966856 |
| 8 | $\log(500)$ | 4.601800 | 1.156325 | 0.974569 |
| 8 | $\log(1000)$ | 4.431683 | 1.069020 | 0.967847 |
| 9 | $\log(500)$ | 5.302598 | 2.415647 | 0.973016 |
| 9 | $\log(1000)$ | 4.027324 | 1.866802 | 0.958805 |
| 10 | $\log(500)$ | 3.224910 | 1.169699 | 0.963358 |
| 10 | $\log(1000)$ | 3.433624 | 0.999967 | 0.949972 |

Table 1: RLCT estimates for `detectA`.

For the problem `detectA` given in Example 2.5 the dimension of parameter space is $\dim W = 30$. We use generalized least squares to fit the RLCT $\lambda$ (with goodness-of-fit measured by $R^2$), the algorithm of which is given in Appendix A. Our results are displayed in Table 1 and Figure 4. Our purpose in these experiments is not to provide high accuracy estimates of the RLCT, as these would require much longer Markov chains. Instead we demonstrate how rough estimates consistent with the theory can be obtained at low computational cost. If this model were regular the RLCT would be $\dim W/2 = 15$.

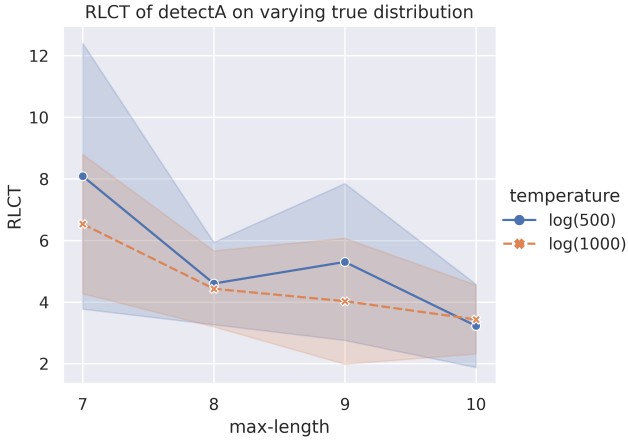

Figure 4: Plot of RLCT estimates for `detectA`. Shaded region shows one standard deviation.

The deterministic synthesis problem `parityCheck` has

$$\Sigma = \{\square, A, B, X\}$$
$$Q = \{\text{reject}, \text{accept}, \text{getNextAB}, \text{getNextA}, \text{getNextB}, \text{gotoStart}\}.$$

The distribution $q(x)$ is as discussed in Section 5 and $q(y|x)$ is determined by the function taking in a string of $A$'s and $B$'s, and terminating in state $\mathrm{accept}$ if the string contains the same number of $A$'s as $B$'s, and terminating in state $\mathrm{reject}$ otherwise. The string is assumed to contain no blank symbols. The true distribution is realisable because there is a Turing machine using $\Sigma$ and $Q$ which computes this function: the machine works by repeatedly overwriting pairs consisting of a single $A$ and $B$ with $X$'s; if there are any $A$'s without a matching $B$ left over (or vice versa), we reject, otherwise we accept.

In more detail, the starting state $\mathrm{getNextAB}$ moves right on the tape until the first $A$ or $B$ is found, and overwrites it with an $X$. If it's an $A$ (resp. $B$) we enter state $\mathrm{getNextB}$ (resp. $\mathrm{getNextA}$). If no $A$ or $B$ is found, we enter the state $\mathrm{accept}$. The state $\mathrm{getNextA}$ (resp. $\mathrm{getNextB}$) moves right until an $A$ (resp. $B$) is found, overwrites it with an $X$ and enters state $\mathrm{gotoStart}$ which moves left until a blank symbol is found (resetting the machine to the left end of the tape). If no $A$'s (resp. $B$'s) were left on the tape, we enter state $\mathrm{reject}$. The dimension of the parameter space is $\dim W = 240$. If this model were regular, the RLCT would be $\dim W/2 = 120$. Our RLCT estimates are contained in Table 2.

| Max-length | Temperature | RLCT | Std | R squared |
|:---:|:---:|:---:|:---:|:---:|
| 5 | $\log(300)$ | 4.411732 | 0.252458 | 0.969500 |
| 6 | $\log(300)$ | 4.005667 | 0.365855 | 0.971619 |
| 7 | $\log(300)$ | 3.887679 | 0.276337 | 0.973716 |

Table 2: RLCT estimates for `parityCheck`.

## 6 DISCUSSION

We have developed a theoretical framework in which *all* programs can in principle be learnt from input-output examples via an existing optimisation procedure. This is done by associating to each program a smooth relaxation which, based on Clift & Murfet (2018), can be argued to be more canonical than existing approaches. This realization has important implications for the building of intelligent systems.

In approaches to program synthesis based on gradient descent there is a tendency to think of solutions to the synthesis problem as isolated critical points of the loss function $K$, but this is a false intuition based on regular models. Since neural networks, Bayesian networks, smooth relaxations of UTMs and all other extant approaches to smooth program synthesis are strictly singular models (the map from parameters to functions is not injective) the set $W_0$ of parameters $w$ with $K(w) = 0$ is a complex extended object, whose geometry is shown by Watanabe's singular learning theory to be deeply related to the learning process. We have examined this geometry in several specific examples and shown how to think about complexity of programs from a geometric perspective. It is our hope that algebraic geometry can assist in developing the next generation of synthesis machines.

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

## APPENDIX

## A    ALGORITHM FOR ESTIMATING RLCTS

Given a sample $D_n = \{(x_i, y_i)\}_{i=1}^n$ from $q(x,y)$ let $L_n(w) := -\frac{1}{n}\sum_{i=1}^n \log p(y_i|x_i, w)$ be the negative log likelihood. We would like to estimate

$$\mathbb{E}_w^\beta[nL_n(w)] := \frac{1}{Z_n^\beta}\int nL_n(w)\varphi(w)\prod_{i=1}^n p(y_i|x_i, w)^\beta dw$$

where $Z_n^\beta = \int \varphi(w)\prod_{i=1}^n p(y_i|x_i, w)^\beta dw$ for some inverse temperature $\beta$. If $\beta = \frac{\beta_0}{\log n}$ for some constant $\beta_0$, then by Theorem 4 of Watanabe (2013),

$$\mathbb{E}_w^\beta[nL_n(w)] = nL_n(w_0) + \frac{\lambda \log n}{\beta_0} + U_n\sqrt{\frac{\lambda \log n}{2\beta_0}} + O_p(1) \tag{7}$$

where $\{U_n\}$ is a sequence of random variables satisfying $\mathbb{E}[U_n] = 0$ and $\lambda$ is the RLCT. In practice, the last two terms often vary negligibly with $1/\beta$ and so $\mathbb{E}_w^\beta[nL_n(w)]$ approximates a linear function of $1/\beta$ with slope $\lambda$ (Watanabe, 2013, Corollary 3). This is the foundation of the RLCT estimation procedure found in Algorithm 1 which is used in our experiments.

---

**Algorithm 1** RLCT estimation

---

**Input:** range of $\beta$'s, set of training sets $\mathcal{T}$ each of size $n$, approximate samples $\{w_1, \ldots, w_R\}$ from $p^\beta(w|\mathcal{D}_n)$ for each training set $D_n$ and each $\beta$
**for** training set $D_n \in \mathcal{T}$ **do**
    **for** $\beta$ in range of $\beta$'s **do**
        Approximate $\mathbb{E}_w^\beta[nL_n(w)]$ with $\frac{1}{R}\sum_{i=1}^R nL_n(w_r)$ where $w_1, \ldots, w_R$ are approximate samples from $p^\beta(w|D_n)$
    **end for**
    Perform generalised least squares to fit $\lambda$ in Equation (7), call result $\hat{\lambda}(D_n)$
**end for**
**Output:** $\frac{1}{|\mathcal{T}|}\sum_{D_n \in \mathcal{T}} \hat{\lambda}(D_n)$

---

Each RLCT estimate $\hat{\lambda}(\mathcal{D}_n)$ in Algorithm 1 was performed by linear regression on the pairs $\{(1/\beta_i, \mathbb{E}_w^{\beta_i}[nL_n(w)])\}_{i=1}^5$ where the five inverse temperatures $\beta_i$ are centered on the inverse temperature $1/T$ where $T$ is the temperature reported for each experiment in Table 1 and Table 2.

From a Bayesian perspective, predictions about outputs $y$ should be made using the predictive distribution

$$p^*(y|x, D_n) = \int p(y|x, w)p(w|D_n)dw \,.$$

The *Bayesian generalisation error* associated to the Bayesian predictor is defined as the Kullback-Leibler distance to the true conditional distribution

$$B_g(n) := D_{KL}(q\|p^*) = \int q(y|x)q(x) \log \left( \frac{q(y|x)}{p^*(y|x)} \right) dy dx \,.$$

If some fundamental conditions are satisfied (Definition 6.1 and Definition 6.3 of Watanabe (2009)), then by Theorem 6.8 of *loc.cit.*, there exists a random variable $B_g^*$ such that as $n \to \infty$, $\mathbb{E}[nB_g(n)]$ converges to $\mathbb{E}[B_g^*]$. In particular, by Theorem 6.10 of Watanabe (2009), $\mathbb{E}[B_g^*] = \lambda$.

## B  HYPERPARAMETERS

The hyperparameters for the various synthesis tasks are contained in Table 3. The *number of samples* is $R$ in Algorithm 1 and the *number of datasets* is $|\mathcal{T}|$. Samples are taken according to the Dirichlet distribution, a probability distribution over the simplex, which is controlled by the *concentration*. When the concentration is a constant across all dimensions, as is assumed here, this corresponds to a density which is symmetric about the uniform probability mass function occurring in the centre of the simplex. The value $\alpha = 1.0$ corresponds to the uniform distribution over the simplex. Finally, the *chain temperature* controls the default $\beta$ value, ie. all inverse temperature values are centered around $1/T$ where $T$ is the chain temperature.

| Hyperparameter | `detectA` | `parityCheck` |
|---|---|---|
| Dataset size ($n$) | 200 | 100 |
| Minimum sequence length ($a$) | 4 | 1 |
| Maximum sequence length ($b$) | 7/8/9/10 | 5/6/7 |
| Number of samples ($R$) | 20,000 | 2,000 |
| Number of burn-in steps | 1,000 | 500 |
| Number of datasets ($|\mathcal{T}|$) | 4 | 3 |
| Target accept probability | 0.8 | 0.8 |
| Concentration ($\alpha$) | 1.0 | 1.0 |
| Chain temperature ($T$) | $\log(500)/\log(1000)$ | $\log(300)$ |
| Number of timesteps ($t$) | 10 | 42 |

Table 3: Hyperparameters for Datasets and MCMC.

## C  THE SHIFT MACHINE

The pseudo-UTM $\mathcal{U}$ is a complicated Turing machine, and the models $p(y|x, w)$ of Section 2 are therefore not easy to analyse by hand. To illustrate the kind of geometry that appears, we study the simple Turing machine `shiftMachine` of Clift & Murfet (2018) and formulate an associated statistical learning problem. The tape alphabet is $\Sigma = \{\Box, A, B, 0, 1, 2\}$ and the input to the machine will be a string of the form $\Box na_1a_2a_3\Box$ where $n$ is called the *counter* and $a_i \in \{A, B\}$. The transition function, given in *loc.cit.*, will move the string of $A$'s and $B$'s leftwards by $n$ steps and fill the right hand end of the string with $A$'s, keeping the string length invariant. For example, if $\Box 2BAB\Box$ is the input to $M$, the output will be $\Box 0BAA\Box$.

Set $W = \Delta\{0, 2\} \times \Delta\{A, B\}$ and view $w = (h, k) \in W$ as representing a probability distribution $(1 - h) \cdot 0 + h \cdot 2$ for the counter and $(1 - k) \cdot B + k \cdot A$ for $a_1$. The model is

$$p(y|x = (a_2, a_3), w) = (1 - h)^2 k \cdot A + (1 - h)^2 (1 - k) \cdot B + \sum_{i=2}^3 \binom{2}{i-1} h^{i-1}(1 - h)^{3-i} \cdot a_i \,.$$

This model is derived by propagating uncertainty through `shiftMachine` in the same way that $p(y|x, w)$ is derived from $\Delta \operatorname{step}^t$ in Section 2 by propagating uncertainty through $\mathcal{U}$. We assume that some distribution $q(x)$ over $\{A, B\}^2$ is given.

**Example C.1.** Suppose $q(y|x) = p(y|x, w_0)$ where $w_0 = (1, 1)$. It is easy to see that

$$K(w) = -\frac{1}{4} \sum_{a_2, a_3} \log p\big(y = a_3 | x = (a_2, a_3), w\big) = -\frac{1}{2} \log[g(h, k)]$$

where $g(h, k) = \big((1 - h)^2 k + h^2\big)\big((1 - h)^2(1 - k) + h^2\big)$ is a polynomial in $w$. Hence

$$W_0 = \{(h, k) \in W : g(h, k) = 1\} = \mathbb{V}(g - 1) \cap [0, 1]^2$$

is a semi-algebraic variety, that is, it is defined by polynomial equations and inequalities. Here $\mathbb{V}(h)$ denotes the vanishing locus of a function $h$.

**Example C.2.** Suppose $q(AB) = 1$ and $q(y|x = AB) = \frac{1}{2}A + \frac{1}{2}B$. Then the Kullback-Leibler divergence is $K(h, k) = -\frac{1}{2} \log(4f(1 - f))$ where $f = (1 - h)^2 k + 2h(1 - h)$. Hence $\nabla K = (f - \frac{1}{2})\frac{1}{f(1-f)}\nabla f$. Note that $f$ has no critical points, and so $\nabla K = 0$ at $(h, k) \in (0, 1)^2$ if and only if $f(h, k) = \frac{1}{2}$. Since $K$ is non-negative, any $w \in W_0$ satisfies $\nabla K(w) = 0$ and so

$$W_0 = [0, 1]^2 \cap \mathbb{V}(4f(1 - f) - 1) = [0, 1]^2 \cap \mathbb{V}(f - \tfrac{1}{2})$$

is semi-algebraic. Note that the curve $f = \frac{1}{2}$ is regular while the curve $4f(1 - f) = 1$ is singular and it is the geometry of the singular curve that is related to the behaviour of $K$. This curve is shown in Figure 5. It is straightforward to check that the determinant of the Hessian of $K$ is identically zero on $W_0$, so that every point on $W_0$ is a degenerate critical point of $K$.

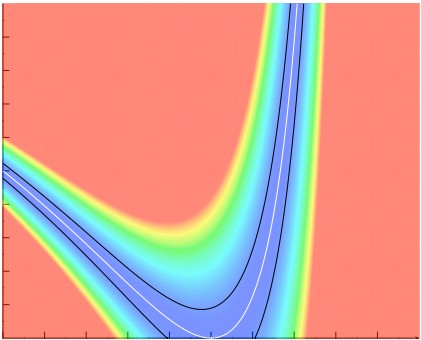

Figure 5: Values of $K(h, k)$ on $[0, 1]^2$ are shown by colour, ranging from blue (zero) to red (0.01). The singular analytic space $K = 0$ (white) and the regular analytic level set $K = 0.001$ (black).

## D    GENERAL SOLUTION FOR DETERMINISTIC SYNTHESIS PROBLEMS

In this section we consider the case of a deterministic synthesis problem $q(x, y)$ which is finitely supported in the sense that there exists a finite set $\mathcal{X} \subseteq \Sigma^*$ such that $q(x) = c$ for all $x \in \mathcal{X}$ and $q(x) = 0$ for all $x \notin \mathcal{X}$. We first need to discuss the coordinates on the parameter space $W$ of (3). To specify a point on $W$ is to specify for each pair $(\sigma, q) \in \Sigma \times Q$ (that is, for each tuple on the description tape) a triple of probability distributions

$$\sum_{\sigma' \in Q} x^{\sigma, q}_{\sigma'} \cdot \sigma' \in \Delta\Sigma \,,$$

$$\sum_{q' \in Q} y^{\sigma, q}_{q'} \cdot q' \in \Delta Q \,,$$

$$\sum_{d \in \{L, S, R\}} z^{\sigma, q}_d \cdot d \in \Delta\{L, S, R\} \,.$$

The space $W$ of distributions is therefore contained in the affine space with coordinate ring

$$R_W = \mathbb{R}\big[\{x^{\sigma,q}_{\sigma'}\}_{\sigma,q,\sigma'}, \{y^{\sigma,q}_{q'}\}_{\sigma,q,q'}, \{z^{\sigma,q}_d\}_{\sigma,q,d}\big].$$

The function $F^x = \Delta\,\mathrm{step}^t(x, -) : W \longrightarrow \Delta Q$ is polynomial (Clift & Murfet, 2018, Proposition 4.2) and we denote for $s \in Q$ by $F^x_s \in R_W$ the polynomial computing the associated component of the function $F^x$. Let $\partial W$ denote the boundary of the manifold with corners $W$, that is, the set of all points on $W$ where at least one of the coordinate functions given above vanishes

$$\partial W = \mathbb{V}\Big(\prod_{\sigma,q}\Big[\prod_{\sigma'\in Q} x^{\sigma,q}_{\sigma'} \prod_{q'\in Q} y^{\sigma,q}_{q'} \prod_{d\in\{L,S,R\}} z^{\sigma,q}_d\Big]\Big)$$

where $\mathbb{V}(h)$ denotes the vanishing locus of $h$.

**Lemma D.1.** $W_0 \neq W$.

*Proof.* Choose $x \in \mathcal{X}$ with $q(x) > 0$ and let $y$ be such that $q(y|x) = 1$. Let $w \in W^{code}$ be the code for the Turing machine which ignores the symbol under the head and current state, transitions to some fixed state $s \neq y$ and stays. Then $w \notin W_0$. $\qquad\square$

**Lemma D.2.** *The set $W_0$ is semi-algebraic and $W_0 \subseteq \partial W$.*

*Proof.* Given $x \in \Sigma^*$ with $q(x) > 0$ we write $y = y(x)$ for the unique state with $q(x, y) \neq 0$. In this notation the Kullback-Leibler divergence is

$$K(w) = \sum_{x\in\mathcal{X}} c\, D_{KL}(y\|F^x(w)) = -c\sum_{x\in\mathcal{X}} \log F^x_y(w) = -c\log\prod_{x\in\mathcal{X}} F^x_y(w).$$

Hence

$$W_0 = W \cap \bigcap_{x\in\mathcal{X}} \mathbb{V}(1 - F^x_y(w))$$

is semi-algebraic.

Recall that the function $\Delta\,\mathrm{step}^t$ is associated to an encoding of the UTM in linear logic by the Sweedler semantics (Clift & Murfet, 2018) and the particular polynomials involved have a form that is determined by the details of that encoding (Clift & Murfet, 2018, Proposition 4.3). From the design of our UTM we obtain positive integers $l_\sigma, m_q, n_d$ for $\sigma \in \Sigma, q \in Q, d \in \{L, S, R\}$ and a function $\pi : \Theta \longrightarrow Q$ where

$$\Theta = \prod_{\sigma,q} \Sigma^{l_\sigma} \times Q^{m_q} \times \{L, S, R\}^{n_d}.$$

We represent elements of $\Theta$ by tuples $(\mu, \zeta, \xi) \in \Theta$ where $\mu(\sigma, q, i) \in \Sigma$ for $\sigma \in \Sigma, q \in Q$ and $1 \le i \le l_\sigma$ and similarly $\zeta(\sigma, q, j) \in Q$ and $\xi(\sigma, q, k) \in \{L, S, R\}$. The polynomial $F^x_s$ is

$$F^x_s = \sum_{(\mu,\zeta,\xi)\in\Theta} \delta(s = \pi(\mu, \zeta, \xi)) \prod_{\sigma,q}\Big[\prod_{i=1}^{l_\sigma} x^{\sigma,q}_{\mu(\sigma,q,i)} \prod_{j=1}^{m_q} y^{\sigma,q}_{\zeta(\sigma,q,j)} \prod_{k=1}^{n_d} z^{\sigma,q}_{\xi(\sigma,q,k)}\Big]$$

where $\delta$ is a Kronecker delta. With this in hand we may compute

$$W_0 = W \cap \bigcap_{x\in\mathcal{X}} \mathbb{V}(1 - F^x_y(w))$$
$$= W \cap \bigcap_{x\in\mathcal{X}} \bigcap_{s\neq y} \mathbb{V}(F^x_s(w)).$$

But $F^x_s$ is a polynomial with non-negative integer coefficients, which takes values in $[0, 1]$ for $w \in W$. Hence it vanishes on $w$ if and only if for each triple $\mu, \zeta, \xi$ with $s = \pi(\mu, \zeta, \xi)$ one or more of the coordinate functions $x^{\sigma,q}_{\mu(\sigma,q,i)}, y^{\sigma,q}_{\zeta(\sigma,q,j)}, z^{\sigma,q}_{\xi(\sigma,q,k)}$ vanishes on $w$.

The desired conclusion follows unless for every $x \in \mathcal{X}$ and $(\mu, \zeta, \xi) \in \Theta$ we have $\pi(\mu, \zeta, \xi) = y$ so that $F^x_s = 0$ for all $s \neq y$. But in this case case $W_0 = W$ which contradicts Lemma D.1. $\qquad\square$

## E    STAGED PSEUDO-UTM

Simulating a Turing machine $M$ with tape alphabet $\Sigma$ and set of states $Q$ on a standard UTM requires the specification of an encoding of $\Sigma$ and $Q$ in the tape alphabet of the UTM. From the point of view of exploring the geometry of program synthesis, this additional complexity is uninteresting and so here we consider a *staged pseudo-UTM* whose alphabet is

$$\Sigma_{\mathrm{UTM}} = \Sigma \cup Q \cup \{L, R, S\} \cup \{X, \square\}$$

where the union is disjoint where $\square$ is the blank symbol (which is distinct from the blank symbol of $M$). Such a machine is capable of simulating any machine with tape alphabet $\Sigma$ and set of states $Q$ but cannot simulate arbitrary machines and is not a UTM in the standard sense. The adjective *staged* refers to the design of the UTM, which we now explain. The set of states is

$$Q_{\mathrm{UTM}} = \{ \text{compSymbol, compState, copySymbol, copyState, copyDir,}$$
$$\neg\text{compState}, \neg\text{copySymbol}, \neg\text{copyState}, \neg\text{copyDir,}$$
$$\text{updateSymbol, updateState, updateDir, resetDescr} \}.$$

The UTM has four tapes numbered from 0 to 3, which we refer to as the *description tape*, the *staging tape*, the *state tape* and the *working tape* respectively. Initially the description tape contains a string of the form

$$X s_0 q_0 s_0' q_0' d_0 s_1 q_1 s_1' q_1' d_1 \dots s_N q_N s_N' q_N' d_N X,$$

corresponding to the tuples which define $M$, with the tape head initially on $s_0$. The staging tape is initially a string $XXX$ with the tape head over the second $X$. The state tape has a single square containing some distribution in $\Delta Q$, corresponding to the initial state of the simulated machine $M$, with the tape head over that square. Each square on the the working tape is some distribution in $\Delta\Sigma$ with only finitely many distributions different from $\square$. The UTM is initialized in state compSymbol.

The operation of the UTM is outlined in Figure 6. It consists of two phases; the *scan phase* (middle and right path), and the *update phase* (left path). During the scan phase, the description tape is scanned from left to right, and the first two squares of each tuple are compared to the contents of the working tape and state tape respectively. If both agree, then the last three symbols of the tuple are written to the staging tape (middle path), otherwise the tuple is ignored (right path). Once the $X$ at the end of the description tape is reached, the UTM begins the update phase, wherein the three symbols on the staging tape are then used to print the new symbol on the working tape, to update the simulated state on the state tape, and to move the working tape head in the appropriate direction. The tape head on the description tape is then reset to the initial $X$.

**Remark E.1.** One could imagine a variant of the UTM which did not include a staging tape, instead performing the actions on the work and state tape directly upon reading the appropriate tuple on the description tape. However, this is problematic when the contents of the state or working tape are distributions, as the exact time-step of the simulated machine can become unsynchronised, increasing entropy. As a simple example, suppose that the contents of the state tape were $0.5q + 0.5p$, and the symbol under the working tape head was $s$. Upon encountering the tuple $sqs'q'R$, the machine would enter a superposition of states corresponding to the tape head having both moved right and not moved, complicating the future behaviour.

We define the *period* of the UTM to be the smallest nonzero time interval taken for the tape head on the description tape to return to the initial $X$, and the machine to reenter the state compSymbol. If the number of tuples on the description tape is $N$, then the period of the UTM is $T = 10N + 5$. Moreover, other than the working tape, the position of the tape heads are $T$-periodic.

## F    SMOOTH TURING MACHINES

Let $\mathcal{U}$ be the staged pseudo-UTM of Appendix E. In defining the model $p(y|x, w)$ associated to a synthesis problem in Section 2 we use a smooth relaxation $\Delta\,\mathrm{step}^t$ of the step function of $\mathcal{U}$. In this appendix we define the smooth relaxation of any Turing machine following Clift & Murfet (2018).

Let $M = (\Sigma, Q, \delta)$ be a Turing machine with a finite set of symbols $\Sigma$, a finite set of states $Q$ and transition function $\delta : \Sigma \times Q \to \Sigma \times Q \times \{-1, 0, 1\}$. We write $\delta_i = \mathrm{proj}_i \circ \delta$ for the $i$th component of $\delta$ for $i \in \{1, 2, 3\}$. For $\square \in \Sigma$, let

$$\Sigma^{\mathbb{Z},\square} = \{ f : \mathbb{Z} \to \Sigma | f(i) = \square \text{ except for finitely many } i \}.$$

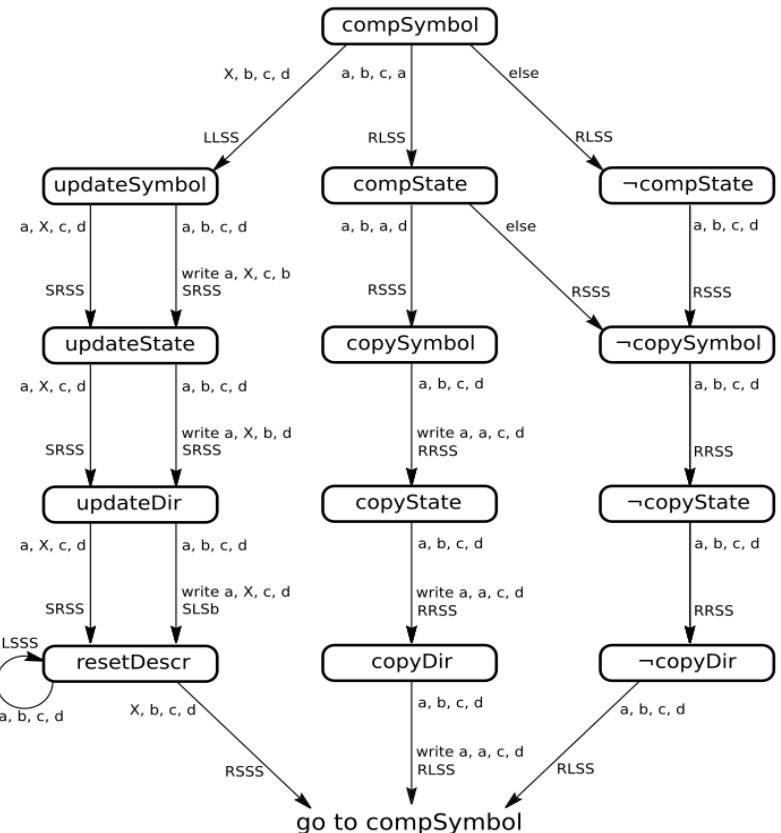

Figure 6: The UTM. Each of the rectangles are states, and an arrow $q \to q'$ has the following interpretation: if the UTM is in state $q$ and sees the tape symbols (on the four tapes) as indicated by the source of the arrow, then the UTM transitions to state $q'$, writes the indicated symbols (or if there is no write instruction, simply rewrites the same symbols back onto the tapes), and performs the indicated movements of each of the tape heads. The symbols $a, b, c, d$ stand for generic symbols which are not $X$.

We can associate to $M$ a discrete dynamical system $\widehat{M} = (\Sigma^{\mathbb{Z}, \square} \times Q, \text{step})$ where

$$\text{step} : \Sigma^{\mathbb{Z}, \square} \times Q \to \Sigma^{\mathbb{Z}, \square} \times Q$$

is the *step function* defined by

$$\text{step}(\sigma, q) = \left( \alpha^{\delta_3(\sigma_0, q)} \left( \ldots, \sigma_{-2}, \sigma_{-1}, \delta_1(\sigma_0, q), \sigma_1, \sigma_2, \ldots \right), \delta_2(\sigma_0, q) \right).$$

with shift map $\alpha^{\delta_3(\sigma_0, q)}(\sigma)_u = \sigma_{u + \delta_3(\sigma_0, q)}$.

Let $X$ be a finite set. The *standard $X$-simplex* is defined as

$$\Delta X = \{ \sum_{x \in X} \lambda_x x \in \mathbb{R} X \mid \sum_x \lambda_x = 1 \text{ and } \lambda_x \geq 0 \text{ for all } x \in X \} \tag{8}$$

where $\mathbb{R} X$ is the free vector space on $X$. We often identify $X$ with the vertices of $\Delta X$ under the canonical inclusion $i : X \to \Delta X$ given by $i(x) = \sum_{x' \in X} \delta_{x = x'} x'$. For example $\{0, 1\} \subset \Delta(\{0, 1\}) \simeq [0, 1]$.

A tape square is said to be at *relative position* $u \in \mathbb{Z}$ if it is labelled $u$ after enumerating all squares in increasing order from left to right such that the square currently under the head is assigned zero. Consider the following random variables at times $t \geq 0$:

- $Y_{u,t} \in \Sigma$: the content of the tape square at relative position $u$ at time $t$.
- $S_t \in Q$: the internal state at time $t$.
- $Wr_t \in \Sigma$: the symbol to be written, in the transition from time $t$ to $t+1$.
- $Mv_t \in \{L, S, R\}$: the direction to move, in the transition from time $t$ to $t+1$.

We call a smooth dynamical system a pair $(A, \phi)$ consisting of a smooth manifold $A$ with corners together with a smooth transformation $\phi : A \to A$.

**Definition F.1.** Let $M = (\Sigma, Q, \delta)$ be a Turing machine. The *smooth relaxation* of $M$ is the smooth dynamical system $((\Delta\Sigma)^{\mathbb{Z},\square} \times \Delta Q, \Delta\text{step})$ where

$$\Delta\text{step} : (\Delta\Sigma)^{\mathbb{Z},\square} \times \Delta Q \to (\Delta\Sigma)^{\mathbb{Z},\square} \times \Delta Q$$

is a smooth transformation sending a state $(\{P(Y_{u,t})\}_{u \in \mathbb{Z}}, P(S_t))$ to $(\{P(Y_{u,t+1})\}_{u \in \mathbb{Z}}, P(S_{t+1}))$ determined by the equations

- $P(Mv_t = d|C) = \sum_{\sigma, q} \delta_{\delta_3(\sigma, q) = d} P(Y_{0,t} = \sigma|C) P(S_t = q|C),$

- $P(Wr_t = \sigma|C) = \sum_{\sigma', q} \delta_{\delta_1(\sigma', q) = \sigma} P(Y_{0,t} = \sigma'|C) P(S_t = q|C),$

- $P(S_{t+1} = q|C) = \sum_{\sigma, q'} \delta_{\delta_2(\sigma, q') = q} P(Y_{0,t} = \sigma|C) P(S_t = q'|C),$

- $\begin{aligned}
P(Y_{u,t+1} = \sigma|C) &= P(Mv_t = L|C)\Big(\delta_{u \neq 1} P(Y_{u-1,t} = \sigma|C) + \delta_{u=1} P(Wr_t = \sigma|C)\Big) \\
&+ P(Mv_t = S|C)\Big(\delta_{u \neq 0} P(Y_{u,t} = \sigma|C) + \delta_{u=0} P(Wr_t = \sigma|C)\Big) \\
&+ P(Mv_t = R|C)\Big(\delta_{u \neq -1} P(Y_{u+1,t} = \sigma|C) + \delta_{u=-1} P(Wr_t = \sigma|C)\Big),
\end{aligned}$

where $C \in (\Delta\Sigma)^{\mathbb{Z},\square} \times \Delta Q$ is an initial state.

We will call the smooth relaxation of a Turing machine a *smooth Turing machine*. A smooth Turing machine encodes uncertainty in the initial configuration of a Turing machine together with an update rule for how to propagate this uncertainty over time. We interpret the smooth step function as updating the state of belief of a "naive" Bayesian observer. This nomenclature comes from the assumption of conditional independence between random variables in our probability functions.

**Remark F.2.** Propagating uncertainty using standard probability leads to a smooth dynamical system which encodes the state evolution of an "ordinary" Bayesian observer of the Turing machine. This requires the calculation of various joint distributions which makes such an extension computationally difficult to work with. Computation aside, the naive probabilistic extension is justified from the point of view of derivatives of algorithms according to the denotational semantics of differential linear logic. See Clift & Murfet (2018) for further details.

We call the smooth extension of a universal Turing machine a *smooth universal Turing machine*. Recall that the staged pseudo-UTM $\mathcal{U}$ has four tapes: the description tape, the staging tape, the state tape and working tape. The smooth relaxation of $\mathcal{U}$ is a smooth dynamical system

$$\Delta\text{step}_{\mathcal{U}} : [(\Delta\Sigma_{\text{UTM}})^{\mathbb{Z},\square}]^4 \times \Delta Q_{\text{UTM}} \to [(\Delta\Sigma_{\text{UTM}})^{\mathbb{Z},\square}]^4 \times \Delta Q_{\text{UTM}}.$$

If we use the staged pseudo-UTM to simulate a Turing machine with tape alphabet $\Sigma \subseteq \Sigma_{\text{UTM}}$ and states $Q \subseteq \Sigma_{\text{UTM}}$ then with some determined initial state the function $\Delta\text{step}$ restricts to

$$\Delta\text{step}_{\mathcal{U}} : (\Delta\Sigma)^{\mathbb{Z},\square} \times W \times \Delta Q \times \mathcal{X} \longrightarrow (\Delta\Sigma)^{\mathbb{Z},\square} \times W \times \Delta Q \times \mathcal{X}$$

where the first factor is the configuration of the work tape, $W$ is as in (3) and

$$\mathcal{X} = [(\Delta\Sigma_{\text{UTM}})^{\mathbb{Z},\square}] \times \Delta Q_{\text{UTM}}$$

where the first factor is the configuration of the staging tape. Since $\mathcal{U}$ is periodic of period $T = 10N + 5$ (Appendix E) the iterated function $(\Delta\text{step}_{\mathcal{U}})^T$ takes an input with staging tape in its

default state $XXX$ and UTM state compSymbol and returns a configuration with the same staging tape and state, but with the configuration of the work tape, description tape and state tape updated by one complete simulation step. That is,

$$(\Delta \operatorname{step}_{\mathcal{U}})^T (x, w, q, XXX, \operatorname{compSymbol}) = (F(x, w, q), XXX, \operatorname{compSymbol})$$

for some smooth function

$$F : (\Delta \Sigma)^{\mathbb{Z}, \square} \times W \times \Delta Q \longrightarrow (\Delta \Sigma)^{\mathbb{Z}, \square} \times W \times \Delta Q . \tag{9}$$

Finally we can define the function $\Delta \operatorname{step}^t$ of (4). We assume all Turing machines are initialised in some common state init $\in Q$.

**Definition F.3.** Given $t \geq 0$ we define $\Delta \operatorname{step}^t : \Sigma^* \times W \longrightarrow \Delta Q$ by

$$\Delta \operatorname{step}^t(x, w) = \Pi_Q F^t(x, w, \operatorname{init})$$

where $\Pi_Q$ is the projection onto $\Delta Q$.

# G  DIRECT SIMULATION

For computational efficiency in our PyTorch implementation of the staged pseudo-UTM we implement $F$ of (9) rather than $\Delta \operatorname{step}_{\mathcal{U}}$. We refer to this as *direction simulation* since it means that we update in one step the state and working tape of the UTM for a full cycle where a cycle consists of $T = 10N + 5$ steps of the UTM.

Let $S(t)$ and $Y_u(t)$ be random variables describing the contents of state tape and working tape in relative positions $0, u$ respectively after $t \geq 0$ time steps of the UTM. We define $\widetilde{S}(t) := S(4 + Tt)$ and $\widetilde{Y}_u(t) := Y_u(4 + Tt)$ where $t \geq 0$ and $u \in \mathbb{Z}$. The task then is to define functions $f, g$ such that

$$\widetilde{S}(t + 1) = f(\widetilde{S}(t))$$

$$\widetilde{Y}_u(t + 1) = g(\widetilde{Y}_u(t)).$$

The functional relationship is given as follows: for $1 \leq i \leq N$ indexing tuples on the description tape, while processing that tuple, the UTM is in a state distribution $\lambda_i \cdot \bar{q} + (1 - \lambda_i) \cdot \neg \bar{q}$ where $\bar{q} \in \{\operatorname{copySymbol}, \operatorname{copyState}, \operatorname{copyDir}\}$. Given the initial state of the description tape, we assume uncertainty about $s', q', d$ only. This determines a map

$$\theta : \{1, \ldots, N\} \to \Sigma \times Q$$

where the description tape at tuple number $i$ is given by $\theta(i)_1 \theta(i)_2 P(s_i') P(q_i') P(d_i)$. We define the conditionally independent joint distribution between $\{\widetilde{Y}_{0,t-1}, \widetilde{S}_{t-1}\}$ by

$$\lambda_i = \sum_{\sigma \in \Sigma} \delta_{\theta(i)_1 = \sigma} P(\widetilde{Y}_{0,t-1} = \sigma) \cdot \sum_{q \in Q} \delta_{\theta(i)_2 = q} P(\widetilde{S}_{t-1} = q)$$

$$= P(\widetilde{Y}_{0,t-1} = \theta(i)_1) \cdot P(\widetilde{S}_{t-1} = \theta(i)_2).$$

We then calculate a recursive set of equations for $0 \leq j \leq N$ describing distributions $P(\hat{s}_j), P(\hat{q}_j)$ and $P(\hat{d}_j)$ on the staging tape after processing all tuples up to and including tuple $j$. These are given by $P(\hat{s}_0) = P(\hat{q}_0) = P(\hat{d}_0) = 1 \cdot X$ and

$$P(\hat{s}_i) = \sum_{\sigma \in \Sigma} \{\lambda_i \cdot P(s_i' = \sigma) + (1 - \lambda_i) \cdot P(\hat{s}_{i-1} = \sigma)\} \cdot \sigma + (1 - \lambda_i) \cdot P(\hat{s}_{i-1} = X) \cdot X$$

$$P(\hat{q}_i) = \sum_{q \in Q} \{\lambda_i \cdot P(q_i' = q) + (1 - \lambda_i) \cdot P(\hat{q}_{i-1} = q)\} \cdot q + (1 - \lambda_i) \cdot P(\hat{q}_{i-1} = X) \cdot X$$

$$P(\hat{d}_i) = \sum_{a \in \{L, R, S\}} \{\lambda_i \cdot P(d_i = a) + (1 - \lambda_i) \cdot P(\hat{d}_{i-1} = a)\} \cdot a + (1 - \lambda_i) \cdot P(\hat{d}_{i-1} = X) \cdot X.$$

Let $A_\sigma = P(\hat{s}_N = X) \cdot P(\widetilde{Y}_{0,t-1} = \sigma) + P(\hat{s}_N = \sigma)$. In terms of the above distributions

$$P(\widetilde{S}_t) = \sum_{q \in Q} \left( P(\hat{q}_N = X) \cdot P(\widetilde{S}_{t-1} = q) + P(\hat{q}_N = q) \right) \cdot q$$

and

$$P(\widetilde{Y}_{u,t} = \sigma) = P(\hat{d}_N = L)\left(\delta_{u \neq 1}P(\widetilde{Y}_{u-1,t-1} = \sigma) + \delta_{u=1}A_\sigma\right)$$
$$+ P(\hat{d}_N = R)\left(\delta_{u \neq -1}P(\widetilde{Y}_{u+1,t-1} = \sigma) + \delta_{u=-1}A_\sigma\right)$$
$$+ P(\hat{d}_N = S)\left(\delta_{u \neq 0}P(\widetilde{Y}_{u,t-1} = \sigma) + \delta_{u=0}A_\sigma\right)$$
$$+ P(\hat{d}_N = X)\left(\delta_{u \neq 0}P(\widetilde{Y}_{u,t-1} = \sigma) + \delta_{u=0}A_\sigma\right).$$

Using these equations, we can state efficient update rules for the staging tape. We have

$$P(\hat{s}_N = X) = \prod_{j=1}^{N}(1 - \lambda_j), \qquad P(\hat{s}_N = \sigma) = \sum_{j=1}^{N}\lambda_j \cdot P(s'_j = \sigma)\prod_{l=j+1}^{N}(1 - \lambda_l)$$

$$P(\hat{q}_N = X) = \prod_{j=1}^{N}(1 - \lambda_j), \qquad P(\hat{q}_N = q) = \sum_{j=1}^{N}\lambda_j \cdot P(q'_j = q)\prod_{l=j+1}^{N}(1 - \lambda_l)$$

$$P(\hat{d}_N = X) = \prod_{j=1}^{N}(1 - \lambda_j), \qquad P(\hat{d}_N = a) = \sum_{j=1}^{N}\lambda_j \cdot P(d_j = a)\prod_{l=j+1}^{N}(1 - \lambda_l).$$

To enable efficient computation, we can express these equations using tensor calculus. Let $\lambda = (\lambda_1, \ldots, \lambda_N) \in \mathbb{R}^N$. We view

$$\theta : \mathbb{R}^N \to \mathbb{R}\Sigma \otimes \mathbb{R}Q$$

as a tensor and so $\theta = \sum_{i=1}^{N} i \otimes \theta(i)_1 \otimes \theta(i)_2 \in \mathbb{R}^N \otimes \mathbb{R}\Sigma \otimes \mathbb{R}Q$. Then

$$\theta \lrcorner \left(P(\widetilde{Y}_{0,t-1}) \otimes P(\widetilde{S}_{t-1})\right) = \sum_{i=1}^{N} i \cdot P(\widetilde{Y}_{0,t-1} = \theta(i)_1) \cdot P(\widetilde{S}_{t-1} = \theta(i)_2) = \lambda.$$

If we view $P(s'_* = \bullet) \in \mathbb{R}^N \otimes \mathbb{R}^\Sigma$ as a tensor, then

$$P(\hat{s}_N) = \sum_{j=1}^{N} P(s'_j = \bullet)\cdot\left(\lambda_j\prod_{l=j+1}^{N}(1-\lambda_l)\right) = \lambda\cdot\left(\prod_{l=2}^{N}(1-\lambda_l),\prod_{l=3}^{N}(1-\lambda_l),\ldots,(1-\lambda_N),1\right)$$

can be expressed in terms on the vector $\lambda$ only. Similarly, $P(q'_* = \bullet) \in \mathbb{R}^N \otimes \mathbb{R}^Q$ with

$$P(\hat{q}_N) = \sum_{j=1}^{N} P(q'_j = \bullet)\cdot\left(\lambda_j\prod_{l=j+1}^{N}(1-\lambda_l)\right) = \lambda\cdot\left(\prod_{l=2}^{N}(1-\lambda_l),\prod_{l=3}^{N}(1-\lambda_l),\ldots,(1-\lambda_N),1\right)$$

and $P(d_* = \bullet) \in \mathbb{R}^N \otimes \mathbb{R}^3$ with

$$P(\hat{d}_N) = \sum_{j=1}^{N} P(d_j = \bullet)\cdot\left(\lambda_j\prod_{l=j+1}^{N}(1-\lambda_l)\right) = \lambda\cdot\left(\prod_{l=2}^{N}(1-\lambda_l),\prod_{l=3}^{N}(1-\lambda_l),\ldots,(1-\lambda_N),1\right).$$

