# OpenReview forum: "Geometry of Program Synthesis"
_ICLR.cc/2021/Conference — Reject_

### Official Review · AnonReviewer4 · 2020-10-23
**Classically, we think of synthesis (by gradient, over a smooth relaxation) as getting lucky of arriving at a solution s.t. the loss is 0. This work says that instead, it is easy to arrive at a point where the KL 0, but it might not be a realizable program.**

**Rating:** 7
**Confidence:** 2

**Review:**

- quality : good
- clarity : very good. I was worried about reading this due to all the math symbols, but it turns out the big pictures are clearly explained.
- originality : okay
- significance : not sure

I could not follow the more technical aspects as I'm not a theory person (although I did fail algebraic geometry at one point). The main claim of the paper is that when doing relaxations and attempt to view satisfying programs as the variety W0 of some mathematical function, W0 is typically larger than W0 \intersect Wcode. This is important because one may easily find a point of solution in the variety W0, yet it cannot be realized as a working program in Wcode. I can confirm that this is indeed the case in some limited capacity when I tried to approximate deterministic functions via neural networks, and attempted to synthesize these functions by "pulling back" from the output to the space of programs, which resulted in a relaxed representation of program that has very low loss, yet do not correspond to actual programs. It is nice to see why this happened in a formal setting.

As I am confident that my expertise in this area is not high, I will not be providing an explicit pro/con list. I would instead ask a few questions to the authors and hope I can get a good response:

- A natural thing to make program synthesis more amendable is by changing the set Wcode. For instance, if I were to represent the function that adds 3, I can either do that with x + 3, or with x+1+1+1. Now, which one of these representation would lead to a Wcode that has a better chance of intersecting better with W0? As DSL designers, we have fairly big freedom in designing what Wcode is, that is often the only thing we can control. Is there a rule of thumb that we can follow to ensure that gradient-based or sampling-based relaxation works well? i.e. W0 isn't such a horrible extended object?

- Another "knob" that we can control easily in synthesis is the number of input output examples, and which examples to give. If I want to specify a polynomial of degree 2, I can choose either 3 points, or 30 points. Which is easier? Specifically, given a specific Wcode, given two dataset of input-output examples, D1 and D2. Let Sol(D,Wcode) denote the set of programs in Wcode that satisfies D completely, in a classical sense. The question is, if Sol(D1, Wcode) = Sol(D2, Wcode), how do we quantify if synthesis is easier over D1 or over D2 ? i.e. which set, Variety(D1) or Variety(D2) is the more horrible extended object of Wcode?

final recommendation:
I maintain my score. I think this is an interesting piece of work that can easily be used as citations whenever the discussion of "why don't you just relax your program to be differentiable" comes up, and I can cite this paper and say "no that does not work theoretically".

---

> ### Author Response · Authors · 2020-11-16
> **Response to AnonReviewer4**
>
> Thanks for the positive comments. You raise several interesting questions.
>
> Firstly, since $W_0$ is higher dimensional than $W^{code} \cap W_0$ it should be easier to find a point of the former than the latter (by either SGD or MCMC). Perhaps one insight that is gained from our point of view is that $W_0$ is an extended object, and while we don’t see how to guarantee it currently in general, if you can find a point on $W_0$ you could reasonably hope to then find a path within $W_0$ to a point of $W^{code} \cap W_0$. So it probably makes sense to divide synthesis into two phases (a) find a point of $W_0$ and (b) retract from $W_0$ to $W^{code} \cap W_0$.
>
> Regarding the design of domain specific languages. In formulating a response to your question we came up with a new insight that now forms Section 4 of the paper. In fact there is a relationship between DSLs and how well the learning process works. I would guess that adding additional higher-level operations to a DSL would decrease the RLCT and thus make the optimisation process work better, but that would require some follow-up work. I think the problem with continuous program synthesis is not that $W_0$ is horrible and extended (for instance deep learning works very well and there the set of true parameters $W_0$ is even more horrible and more extended) but rather that the RLCT is too big ($W_0$ is not singular enough) and the number of training examples is generally too small. It could be hoped that a careful study of DSLs in the setting of singular learning theory might shed some light on this.
>
> Regarding the number of training examples. Again, this question prompted some of the discussion now in Section 4, which clarifies that generically you expect that the more examples you provide the less likely the optimisation process will get stuck in a local minimum of the free energy.

---

> > ### Comment · AnonReviewer4 · 2020-11-16
> > **thanks for the responses :)**
> >
> > thanks for the responses, they have been helpful.
> >
> > one point that's interesting is the suggestion to "first find a solution in W0 then retract it to W0 \intsersect Wcode. There seems to be a connection between this approach and the A* algorithm, where the solution in W0 can serve as a heuristic for whether a solution exists or not. It is also interesting to see whether we can take W0 and study what kinds of properties it has, and use these properties to guide the search in Wcode.

---

### Official Review · AnonReviewer1 · 2020-10-28
**Very hard to understand**

**Rating:** 5
**Confidence:** 1

**Review:**

This paper is very densely written with a lot of heavy mathematical formalism and not enough schematics to explain how it works. Most of the foundations are coming from theoretical computer science that is beyond my area of expertise. I spent many hours and still don't understand the point that the authors are trying to make. It seems to me that they are trying to use some extensions of the manifold variety in order to do program synthesis. They support their thesis with some theorems and assumptions around the geometry of program synthesis. The whole paper revolves around the zeros of the KL divergence between q(x,y) and p(y|x,w). They give a detailed analysis of these functions and justify why the Singular Learning Theory is appropriate for this study.
Following the references wasn't easy. The paper is based on this work https://arxiv.org/pdf/1805.11813.pdf which is outside of my research area.

I would suggest a revision and a lighter submission of the paper, I am not sure it is appropriate for this venue

---

> ### Author Response · Authors · 2020-11-16
> **Response to AnonReviewer1**
>
> Thanks for the comments, we have added some remarks to the introduction that hopefully clarify what points the paper is trying to make. In short, we explain a geometric point of view on continuous program synthesis and how this (a) explains how to think about classical concepts like Kolmogorov complexity in the context of gradient-based or sampling-based program synthesis and (b) allows for a novel point of view on some of the most pressing problems in program synthesis.
>
> The revised submission is now lighter with some of the more technical contributions moved to the appendices. We would also like to clarify that whilst our smooth extension of Turing machines is based on https://arxiv.org/pdf/1805.11813.pdf, the relevant definition from that paper is contained in Appendix F using more familiar language. This makes our paper self-contained with respect to that reference.
>
> The reviewer comments prompted us to work harder to explain the kind of practical insights that can be derived once program synthesis is placed on a clear mathematical footing within singular learning theory (see the new Section 4).

---

### Official Review · AnonReviewer2 · 2020-11-01
**Review for "Geometry of Program Synthesis"**

**Rating:** 4
**Confidence:** 1

**Review:**

The authors apply algebraic geometry to program synthesis, by identifying programs with points of analytic varieties. They construct a smooth relaxation to the synthesis problem by considering the space of probability distributions over codes for universal turning machines (which is a smooth/continuous manifold), and then translate this probability into corresponding probabilities of generating a correct program. This allows them to extend the KL divergence on the discrete distribution of codes to a smooth function, whose zeros correspond to the desired programs. They use MCMC to find (approximate) these zeros.

Overall, the approach seems interesting, but seems to be a bit complex for a real application (both examples considered in the experiments seem pretty small/toy problems). While the ideas are interesting, it seems like more work needs to be done before they could be of interest to practical program synthesis (admittedly, the authors do not claim that it is practical, but I am unable to judge the paper by quality of the theoretical work)

---

> ### Author Response · Authors · 2020-11-16
> **Response to AnonReviewer2**
>
> Thanks for the comments. We understand your concerns regarding the applicability of our approach. We have developed both a new method which can synthesise arbitrary programs, and the theoretical framework around which to understand the loss surface and model complexity of these programs. This involves considerable mathematical machinery, which is understandable since it is well known that the set of critical points of any loss function does not generically have a simple manifold structure. Without such precise mathematical language, one cannot hope to even talk about solutions (ie. critical points or singular points) to program synthesis tasks.
>
> The first step in developing a new approach to a problem, be it program synthesis or otherwise, is to show in detail that it works on simple examples. Scaling to larger problems is a second step. Incidentally, the whole field of program synthesis is still grappling with small/toy problems which motivated this work. We have included a new Section 4 which gives some practical implications of our method. We have also provided various updates to help improve the clarity of the paper.
>
> We would be happy to discuss any further concerns regarding our approach.

---

### Author Response · Authors · 2020-11-16
**Revision #1**

Thank you to all reviewers for their comments. We have uploaded a new revision, which
- further clarifies the overall purpose of the paper in the introduction.
- rearranges some of the content for clarity.
- adds a new Section 4 on “Practical implications”.

In our opinion, while continuous program synthesis continues to make incremental advances, without bold new ideas it will fall short of its potential. We have contributed such ideas, and in the new Section 4 we sketch just one of the many lines of investigation that are thereby opened up (namely, the role of “choice of DSL” in determining the ease of optimisation). In the original submission we did not make a sufficient effort to explain the significance of our work and its potential for future practical impact, and we thank the reviewers for prompting us to correct this.

---

### Decision · Program_Chairs · 2021-01-07
**Final Decision**

**Decision:**

Reject

**Comment:**

This paper is a bad fit for ICLR and the authors may consider submitting to more theoretical venues. This paper studies algebraic geometry (an area unfamiliar to most ICLR readers) of program synthesis, with the "hope that algebraic geometry can assist in developing the next generation of synthesis machines." Unfortunately, this paper does not get far enough down that path, and its implications cannot realistically be appreciated by an ICLR audience. The reviewers indicate that their low confidence is due to their lack of understanding of algebraic geometry and not due to their lack of understanding of program synthesis. The featured implication of the paper is that synthesized programs are singularities of analytic functions, which is not very meaningful to the ICLR audience. Even if external reviewers verified the correctness of the math, the ICLR audience would still not understand the implications.